# Varietal Influence on the Formation of Bioactive Amines during the Processing of Fermented Cocoa with Different Pulp Contents

**DOI:** 10.3390/foods12030495

**Published:** 2023-01-20

**Authors:** Paulo Túlio de Souza Silveira, Maria Beatriz Abreu Glória, Isabela Portelinha Tonin, Marina Oliveira Paraíso Martins, Priscilla Efraim

**Affiliations:** 1School of Food Engineering, Universidade Estadual de Campinas, Campinas 13083-970, Brazil; 2School of Pharmacy, Universidade Federal de Minas Gerais, Belo Horizonte 31270-901, Brazil; 3Agrícola Conduru, Ilhéus 45666-000, Brazil

**Keywords:** amine, fermentation, PS 1319, Parazinho, cocoa pulp, chocolate

## Abstract

During cocoa processing, there can be the formation of bioactive amines, which are compounds that play relevant roles not only in plant development but also in human health. Thus, we aimed to investigate the presence and levels of bioactive amines during the processing of two important varieties of cocoa (PS 1319 and Parazinho). The seeds were fermented using five different pulp proportions: 100% (E1), 80% (E2), 60% (E3), and 0% (total pulp removal) (E4). The beans were fermented and dried on a farm following traditional procedures. Soon after, they were roasted and processed into chocolates with 60% cocoa in the laboratory. Bioactive amine contents were determined by ion-pair reversed-phase HPLC and fluorometric detection in the samples before, during, and after fermentation, after drying and roasting (nibs), and in the liquor and chocolate. The only amines found before processing in PS 1319 and Parazinho, respectively, in dry weight basis (dwb), were putrescine (pulp, 13.77 and 12.31; seed, 5.88 and 4.58) and serotonin (seed, 2.70 and 2.54). Fermentation was shorter for Parazinho (156 h) compared to PS 1319 (180 h). The changes in amines were affected by the cocoa variety. During drying, the presence of cadaverine stood out, appearing in all treatments of the PS 1319 variety, reaching 17.96 mg/kg dwb, and in two treatments of the Parazinho variety (100 and 60% pulp). During roasting, most of the amines decreased, except for phenylethylamine, which increased up to 2.47 mg/kg dwb for Parazinho and 1.73 mg/kg dwb for PS 1319. Most of the amines formed and built up (e.g., tyramine, putrescine, and cadaverine) during fermentation were not available or were at low levels in the nibs. Most of the amines found during processing did not reach the final product (chocolate), except for cadaverine in PS 1319 without pulp (7.54 mg/kg dwb). Finally, we confirmed how pulp content, processing, and variety influence the content of bioactive amines in cocoa and chocolate. These changes can be better demonstrated through a heatmap and principal component analysis.

## 1. Introduction

Bioactive amines are ammonia-derived low-molecular-mass organic bases. They are formed when alkyl or aryl groups replace one, two, or three hydrogen atoms to generate primary, secondary, and tertiary amines, respectively. Amine formation occurs in the normal metabolic processes of microorganisms, plants, and animals, which also generate them in food [1,2]. The types and quantity of amines in food depend on its nature, origin, the microbiota (inherent, added, or contaminant), the presence of precursors, e.g., free amino acids, amino acid decarboxylating enzymes, and favorable conditions for the formation of amines [3,4,5,6].

Some amines can occur naturally in plants, playing important roles as hormone precursors and in protecting plants from predators. However, amines can also be formed and accumulate in foods during processing. During drastic thermal processing, including roasting, there can be thermal decarboxylation of free amino acids, producing amines. During fermentation, there can be decarboxylation of free amino acids, especially via microbial decarboxylating enzymes [1,7]. Furthermore, contaminating microorganisms can also produce amines, including putrescine and cadaverine, which, along with other amines, may serve as an index of the hygienic and sanitary conditions prevalent during processing [1,6,8]. 

During cocoa fermentation, several complex biochemical reactions occur simultaneously. The mucilage (pulp) surrounding the seeds, which is rich in carbohydrates, is susceptible to degradation, generating acids by the inherent microbiota [9]. The high acidity and heat generated during fermentation activate endogenous proteolytic enzymes, which hydrolyze proteins into free amino acids and peptides. These compounds, along with reducing sugars, are necessary for the Maillard reaction during drying and roasting, generating desirable chocolate flavor [9]. The increased acidity also activates microbial amino acid decarboxylases, which are responsible for the formation of bioactive amines [9]. Therefore, amine concentration stems from the balance between the formation and degradation of these compounds due to complex biochemical reactions and microorganism diversity. During the fermentation of Bahia cocoa, the levels of serotonin and tryptamine decreased, whereas phenylethylamine increased [9]. The accumulation of this amine is relevant, as it is a neuroactive amine that stimulates the hypothalamus, releases catecholamine, and induces pleasurable sensations, modulating mood and sexual drive [10]. During the fermentation of Pará cocoa, tryptamine, tyramine, spermidine, and spermine were detected. There was a decrease in the levels of tryptamine and an increase in the levels of the polyamines spermine and spermidine [11].

The presence of amines in chocolate and food in general may have health benefits. For example, low concentrations of phenylethylamine are associated with improved cognitive functions and memory, the prevention of diseases such as depression and schizophrenia, and pleasant sensations. The polyamines spermidine and spermine have antioxidant activity, preventing DNA and cell membrane damage. Tyramine and histamine are neurotransmitters. The former has antioxidant and anti-inflammatory properties; the latter, vasodilating ones. However, some amines at high concentrations may cause adverse effects on human health. For example, at high levels, histamine can lead to hypotension, erythema, headache, allergy, and anaphylactic shock, and tyramine can cause hypertension and hypertensive crises [1,12,13,14,15,16,17,18,19].

Recent studies [9,11,20] showed that cocoa processing produces amines, especially during fermentation. However, few studies have correlated the influence of the quantity of pulp on amine content during processing and in the main final product: chocolate. Thus, this study aimed to investigate the formation and presence of bioactive amines during the processing of two widely used varieties of cocoa (PS 1319 and Parazinho) with different proportions of pulp to obtain chocolates.

## 2. Material and Methods

### 2.1. Material

Parazinho (a Brazilian *Forastero* amelonado variety) and PS 1319 (a Brazilian complex hybrid) were used, which were kindly supplied in December 2018 by Fazenda Luz do Vale (14°23′17.2″ S, 39°19′48.4″ W), part of Agrícola Conduru, in Ilhéus, Southern Bahia, Brazil [21]. Both selected varieties are widely grown in Bahia and Espirito Santos states in Brazil and in other cocoa-producing regions. The first one (Parazinho) is not only grown throughout Brazil but also in several other producing countries. 

The reagents used were analytical grade, except for the HPLC ones (LC grade). Ultrapure water was from Milli-Q Plus (Millipore Corp., Milford, MA, USA). The amine derivatization agent (*o*-phthalaldehyde) and the amines (putrescine dihydrochloride, spermidine trihydrochloride, spermidine tetrahydrochloride, agmatine sulfate, cadaverine dihydrochloride, serotonin hydrochloride, histamine dihydrochloride, tyramine hydrochloride, tryptamine, and 2-phenylethylamine hydrochloride) were acquired from Sigma-Aldrich Chemical Co. (St. Louis, MO, USA). 

### 2.2. Cocoa and Chocolate Processing 

The cocoa pods were harvested at the appropriate stage of maturation, and after three days, the pods were cut open with stainless steel knives. A fruit pulper (Model NB10, Bonina, Itabuna, BA, Brazil) was used to remove the pulp from the seed. Separated samples of pulp and seed were taken and lyophilized (Model Alpha 2-4 LD, Christ, Germany) for characterization regarding bioactive amines. 

To investigate the influence of the amount of pulp surrounding the seed during fermentation on bioactive amine formation, different pulp percentages per cocoa bean were used, as follows: Experiment 1 (E1): cocoa beans with the original pulp (100%) pulp; Experiment 2 (E2): beans with 80% original pulp; Experiment 3 (E3): beans with 60% original pulp; and Experiment 4 (E4): total pulp removal (0% pulp). Fermentations were carried out in 45 × 45 × 22 cm (L × W × H) wooden boxes, with a 30 kg initial mass covered with banana leaves. The cocoa mass was rotated according to its temperature [22]. The temperature (digital thermometer, Gulterm Model 180, Gulton, SP, Brazil) and the pH (pHmeter Model HI8424, Hanna Instruments, Woonsocket, RI, USA) were monitored during fermentation and used as parameters for ending the fermentation. Every 24 h, throughout fermentation, samples were collected in aseptic packages, which were frozen and stored (−18 °C) for the analysis of free bioactive amines. The end of fermentation was defined based on the evolution of the temperature and pH of the cocoa mass in the boxes and also based on the color and aroma of the fermented cocoa. After fermentation, the cocoa beans were sun dried on wooden drying barges down to a moisture content of approximately 8.0 g/100 g. The fermented and dried cocoa beans and some of their original pods were sent to the School of Food Engineering at the State University of Campinas (FEA/UNICAMP) for further processing. The pods were opened, pulped by a blade pulper (Model M6998, Langsenkamp, Indianopolis, IN, USA), and frozen. 

Prior to chocolate processing, the fermented and dried beans were roasted in a rotary drum roaster (JAF INOX, Tambaú, SP, Brazil) at 120 °C (on the jacket) for 75 min. After that, cocoa beans were broken using a knife mill (ICMA, Sumaré, SP, Brazil), and the shell and germ were removed (Shell separator, Model 25595, CAPCO, Ipswich, Suffolk, UK) for delivery of the nibs. The nibs were refined in a stone mill/melanger (Model 11, Spectra, Coimbatore, TN, India) to obtain liquor with granulometry <30 μm. The liquor was used to manufacture chocolate containing 60% cocoa to which sugar and lecithin were added. The misture was mixed, refined to a granulometry <25 μm, and conched in the same melanger cited above. 

The samples taken for analysis included pulp and beans collected before and during fermentation, fermented and dried beans, roasted cocoa nibs, liquor, and chocolate. The samples were frozen (−18 °C) and taken to the Faculty of Pharmacy at the Universidade Federal de Minas Gerais for analysis of free bioactive amines and to FEA/UNICAMP for moisture analysis. 

### 2.3. Determination of Free Bioactive Amines

Cocoa-typical free bioactive amines (spermidine, putrescine, agmatine, cadaverine, serotonin, histamine, tyramine, tryptamine, and phenylethylamine) were extracted from samples (5 g) with 7 mL of 5% trichloroacetic acid, which were mixed in a shaker for 5 min and centrifuged at 11,180× *g* at 4 °C for 10 min. Samples were extracted twice more, and the supernatants were filtered through qualitative paper and combined into a 25-mL volumetric flask. The amines were determined using a Shimadzu LC-10AD high-performance liquid chromatograph (HPLC) with a fluorescence detector [9]. The system had an automatic SIL-10AD VP injection system (Shimadzu, Kyoto, Japan), and amines were separated using a Novapak C18 column (3.9 × 300 mm, 4 μm, 60 Å, Waters, Milford, MA, USA), an elution gradient of 0.2 mol/L sodium acetate and 15 mmol/L sodium octanesulfonate (pH adjusted to 4.9, mobile phase A), and acetonitrile (phase B). The amines were identified by comparing retention time and co-elution with standards for amines. The amines were quantified by fluorimetry (340 and 445 nm of excitation and emission, respectively), after post-column derivatization with *o*-phthalaldehyde, and interpolated in analytical curves (5 concentrations, 2 replicates, r^2^ > 0.99). The results were expressed in mg/kg of cocoa beans on a dry weight basis (dwb).

### 2.4. Moisture Content

The moisture content was determined by the gravimetric method using an air circulation oven (Model TE394/1, Tecnal, Piracicaba, SP, Brazil), according to AOAC method 970.20 [23].

### 2.5. Data Analysis

The amine results were subjected to a heat map (Microsoft Excel, version 2211, 2022) and multivariate principal component analysis (PCA) via the “factoshiny” package in R with the RStudio interface [24].

## 3. Results

### 3.1. Temperature and pH Monitoring during Fermentation

Overall, fermentation of the PS 1319 variety lasted 180 h, whereas the Parazinho variety fermented for 156 h. The changes in the cocoa mass temperature and pH during fermentation for both varieties included in this study are described in Figure 1. The evolution of these parameters followed what is normally found for cocoa fermentation [9,11]. The temperature and pH changes during fermentation of the two varieties followed different patterns; however, the changes were similar among treatments for each variety. During fermentation of the PS 1319 variety, temperature changes were similar for all four treatments; the temperature increased from 54 h onward, regardless of pulp percentages. All treatments completed fermentation in 162 h, except for the one with 80% pulp (E2), which continued to ferment up to 180 h, when the temperature finally decreased. During fermentation of the Parazinho cocoa, the temperature increased earlier, compared to PS 1319 at 24 h fermentation. However, we observed that it had greater difficulty maintaining its temperature, which decreased from 72 h onward. The treatments completed fermentation at 156 h and had similar temperatures. The temperature behavior observed during fermentation is typical for cocoa, i.e., an initial increase and a subsequent decrease, indicating the evolution of fermentation via yeasts followed by bacteria, producing alcohol and then acids. The increase in temperature is very important as it kills the seed germ (preventing seed germination) and activates proteolytic enzymes, which lead to the formation and accumulation of precursors of the Maillard reaction. Other authors observed a similar behavior during fermentation of Brazilian cocoa, with lower temperatures at the beginning and end of fermentation, but above 40 °C during the middle of fermentation [9,11].

With respect to the pH of the fermenting cocoa mass, PS 1319 had an initial pH of 3.5 prior to fermentation. The pH decreased in the first 12 h and gradually increased during fermentation. After 144 h, there was a new pH decrease, indicating the end of fermentation. The sample with 80% pulp (E2) fermented for a longer time and included a considerable increase in pH in the last 12 h. For Parazinho, the pH changes were different during fermentation. At the beginning of the fermentation, the pH varied from 3.5 to 4.5, it decreased in the first 48 h and gradually increased during the remainder of the process. After 132 h, there was a new pH decrease, indicating the end of fermentation. Similar pH ranges for both varieties have been described in the literature during cocoa fermentation, for example, pH between 4.43 and 6.17 during fermentation of Colombian Criollo cocoa [25].

### 3.2. Amine Contents in Cocoa Pulp and Seeds Prior to Fermentation

Only two amines, putrescine and serotonin, were detected (nd ≤ 0.40 mg/kg) in the cocoa prior to fermentation. Both amines were present in the seeds, whereas only putrescine was present in the pulp. To the best of our knowledge, this is the first report of bioactive amines in cocoa pulp. The levels of putrescine in the pulps were similar (13.77 and 12.31 mg/kg dwb), irrespective of the variety (PS 1319 or Parazinho). The seeds of the cocoa varieties also had similar levels of putrescine (5.83 and 4.58 mg/kg dwb, for PS 1319 and Parazinho, respectively) and serotonin (2.70 and 2.54 mg/kg dwb, for PS 1319 and Parazinho, respectively). Similar profiles and amine contents were reported in cocoa seeds from the same region (Southern Bahia) prior to fermentation [9]. 

### 3.3. Changes in Bioactive Amines throughout Fermentation

Even though serotonin was detected in the cocoa seed prior to fermentation, it was not detected (<0.40 mg/kg) during fermentation, likely due to dilution in the cocoa mass (seed + pulp). On the other hand, the remaining bioactive amines were detected in considerable amounts, especially during fermentation. 

The PS 1319 variety contained putrescine (Figure 2), especially at the beginning of fermentation for the treatment with 60% pulp (E3), at the end of fermentation for E2 (80% pulp), in the treatment with 100% pulp (E1), and at most fermentation stages. Cadaverine was present (Figure 2), especially at the end of the fermentation stage. PS 1319 cocoa contained higher concentrations of this amine compared to Parazinho, whereas the treatment with no pulp (E4) had the lowest levels.

Tyramine was detected (Figure 3) throughout processing, but especially during the fermentation of the PS 1319 variety. Higher levels were found for the E2, E3, and E4 treatments (80, 60, and 0% pulp, respectively) compared to the E1 treatment (100% pulp), showing how pulp removal influences its emergence. The Parazinho variety showed its highest values in the E2 and E3 treatments (80 and 60% pulp, respectively), which had partially removed pulp. 

Histamine (Figure 3) was present only during the fermentation of PS 1319 beans for treatment with 80% pulp (114 h, E2) and 100% pulp (120 h, E1). We only found it in Parazinho for the fermentation treatment with 60% pulp (0 h, E4). These data show its poor resistance to processing.

We only found agmatine (Figure 4) at a few fermentation moments, which also indicates its low resistance to the other processing stages. The PS 1319 variety shows its presence, especially for treatment E3 (60% of pulp). For both varieties, its complete absence occurs in pulpless treatment (E4), indicating its correlation with the absence of pulp. 

We found that spermidine (Figure 4) is the most common amine during fermentation. Both evaluated varieties contained it, especially at the beginning and middle of fermentation, reaching its peak at 72 h of the process, with the main emphasis on the PS 1319 variety. We observed its decrease at the final times of the fermentation process.

When correlated with temperature and pH results (Figure 1), we found that an important phenomena occurred during cocoa fermentation, such as the beginning of bacterial action, which justifies the considerable increase in temperature (especially for PS 1319) and pH (especially for Parazinho) near this period. We found phenylethylamine (Figure 5) at some points during fermentation. The PS 1319 variety showed higher levels, especially during intermediate fermentation periods. Treatment with 100% pulp (E1) stood out, and we found no level of this amine at any time during fermentation of cocoa without pulp (E4). Parazinho showed the absence of this amine only in the 60% pulp treatment (E3). However, other treatments only showed it at one point throughout the process. Still, we failed to find phenylethylamine in the final phase of fermentation for both varieties. Thus, in addition to the influence of variety, process length determines the resistance of this amine. 

Variety clearly influenced tryptamine (Figure 5). PS 1319 showed it only at 120 and 156 h of fermentation. Moreover, only 100% (E1), 80% (E2), and 0% (E4) pulp treatments showed considerable values for this amine. We detected tryptamine at the beginning of the Parazinho fermentation (with 0% pulp—E4) and within 132, 144, and 156 h for the 100% (E1), 80% (E2), and 60% (E3) treatments.

### 3.4. Influence of Drying

Table 1 shows the amine content of fermented and dried cocoa. We found no tyramine, histamine, serotonin, agmatine, or phenylethylamine. We observed, among the amines found (especially putrescine and cadaverine), the influence of cocoa variety. Especially for PS 1319, which had higher levels compared to Parazinho cocoa. We found putrescine in treatments with 100% (E1) and 80% pulp with the PS 1319 variety. We found it in the Parazinho variety under 60% (E3) and 100% (E1) treatments. Its absence in fermented and dried cocoa (and its low incidence) under pulpless (E4) treatments suggests the influence of pulp removal on its resistance. Moreover, variety seems to influence its incidence, since we found much lower values for it. We found cadaverine in PS 1319 treatments, although those with less pulp showed lower amine content: 60% and 0% pulp, E3 and E4, respectively. In Parazinho, cadaverine behaved similarly to putrescine in treatments with 60% (E3) and 100% (E1) pulp. We found no spermidine in PS 1319 cocoa with 60% pulp (E3). The Parazinho variety showed no spermidine only in the treatment with 80% pulp (E2). 

### 3.5. Influence of Roasting

Table 1 also shows the amines content of roasted cocoa (nibs). We found no histamine, serotonin, agmatine or tryptamine. We detected tyramine, putrescine, and spermidine in the PS 1319 variety, but none in the Parazinho one. We found tyramine and spermidine in treatments with 100% (E1) and 80% (E2) pulp. We detected putrescine in all pulp treatments. Cadaverine behaved similarly to putrescine and was found in all pulp treatments of the PS 1319 variety. However, it was present in only one treatment of the roasted Parazinho cocoa: 0% pulp (E4). We observed phenylethylamine in all PS 1319 cocoa treatments. However, as observed for cadaverine, we found it only in the treatment with 0% pulp (E4) of Parazinho. Finally, we found tryptamine exclusively in the E2 treatment (80% pulp) of Parazinho. 

In general, except for tryptamine, the variety, especially PS 1319, influenced the formation of amines in roasted cocoa. Note that, except for the treatments with 0% pulp (E3), which contained cadaverine and phenylethylamine and the treatment with 80% pulp (E2), in which we observed tryptamine, we found no other amine in any treatment of the Parazinho variety. Moreover, pulp content also influenced it, especially in the pulpless treatments, which failed to form four of the six amines we identified in roasted cocoa. Exceptions to this pattern was observed for cadaverine (Parazinho) and phenylethylamine (PS 1319 and Parazinho).

### 3.6. Influence of Liquor Production

Table 2 shows the amine contents of liquor and chocolate, and no histamine, serotonin, agmatine, or tryptamine were found. We detected tyramine, putrescine, and cadaverine in most treatments of PS 1319 and in some with Parazinho. They were, however, absent in the treatments without pulp (0% pulp—E4). This behavior resembles that of roasted cocoa (except for cadaverine in Parazinho) and shows how pulp content influences the formation of these amines.

On the other hand, we found spermidine only in two treatments: 80% pulp (E2) for PS 1319 and 0% pulp (E4) for Parazinho. We had already observed this amine in treatment E2 of roasted PS 1319 cocoa. Phenylethylamine was the most abundant amine at the end of cocoa liquor production, present in all Parazinho treatments and in those with 100% (E1), 80% (E2), and 60% (E3) of the PS 1319 variety. We found tryptamine neither in PS 1319 liquor nor in roasted cocoa (Table 1). Finally, we only found tryptamine in the treatment E2 (80% pulp) of Parazinho.

### 3.7. Influence of Chocolate Production

Table 2 also shows the amine content of chocolates. We found no histamine, serotonin, agmatine, or tryptamine. We found no putrescine and cadaverine in any of the chocolates made with Parazinho. We observed no tyramine in treatments with 100% (E1) and 60% (E3) pulp for PS 1319. We also failed to find it in the treatment with 60% (E3) of Parazinho pulp. Its quantification is important since it is one of the amines most involved in adverse health effects, which, along with tryptamine, can cause migraines [18]. Putrescine and cadaverine showed similar behavior. Almost all treatments contained them—except that with 0% pulp (E4), which showed no putrescine and the lowest cadaverine index. We observed spermidine only in the treatment with 60% (E3) of Parazinho pulp. In addition to the E3 treatment, it was found in 100% (E1) of PS 1319 pulp.

In order to avoid adverse effects on human health upon consumption of the chocolates, the results demonstrating the absence of histamine and low levels of tyramine in chocolates are interesting. These amines were not found in the chocolates produced from cocoa beans with 60% of its initial pulp content. In addition, the presence of phenylethylamine, which acts as a stimulant, in every treatment is an important beneficial agent. The presence of spermidine, which may have a beneficial antioxidant effect and is also found in treatments with 60% pulp, highlights the quality of chocolates from this treatment [11].

### 3.8. Analysis of the Principal Components and Heatmap

The heatmap (Figure 6) demonstrates the incidence of bioactive amines during cocoa and chocolate processing. The figure corroborates the findings that bioactive amines are present at higher concentrations during fermentation, with emphasis on the PS 1319 variety. Tyramine, putrescine, cadaverine, spermidine, and phenylethylamine are also identified in chocolates. On the other hand, other amines that appear in (Figure 6) throughout the process are not present in the final product. The figure also shows the influence of variety, pulp content, and processing on the contents of bioactive amines. 

Multivariate analysis of data from each processing stage indicated a 39.05% variance for PS 1319 and 43.32% for Parazinho (Figure 7). Both varieties differed during fermentation, and especially in roasted beans and chocolate. PS 1319 showed more noticeable differences between fermentation and chocolate processes. The presence of phenylethylamine, tyramine, cadaverine, and putrescine were indispensable to differentiate these groups. Parazinho had a predominance of agmatine, putrescine, and tyramine; the latter two indicating their influence during processing. The main component analysis comparing the varieties (Figure 8) indicated a 33.21% variance, correlating with the data in Figure 7. We found that the presence of cadaverine, putrescine, tyramine, phenylethylamine, and agmatine were vital to both PS 1319 and Parazinho.

## 4. Discussion

The changes in pH and temperature during fermentation are due to a succession of microbial activities and biochemical reactions that take place [26]. Differences associated with the chemical characteristics (sugar content, acidity) of a specific cocoa variety and with the inherent microbiota can affect changes in temperature and pH. According to Pereira et al. [27], two harvests of Parazinho and PS 1319 differed in total soluble solids, acidity, ascorbic acid, and proteins. They also highlighted how cultivation periods affected cocoa characteristics. In addition, PS 1319 was reported to have a higher pulp content (31.1%) than six other varieties. In addition, it also differed significantly with respect to pH, total soluble solids, and total titratable acidity [28]. 

Immediately prior to fermentation, PS 1319 cocoa mass contained only putrescine, cadaverine, and spermidine. Spermidine was prevalent in the treatments with lower percentage of pulp or without pulp (E3 and E4, respectively). Cadaverine was detected at low levels in the treatment without pulp (E4), whereas putrescine was present in the treatments with 100% and 60% pulp (E1 and E3, respectively).

On the other hand, immediately prior to fermentation, the cocoa seeds of Parazinho showed a larger number of amines (7 amines), compared to PS 1319 (3 amines). There was a wide variation in amines in the cocoa mass, however, major differences were observed for spermidine and agmatine in 100% pulp, which had higher levels compared to the others.

Based on these results, the cocoa seeds prior to fermentation showed differences when comparing the two varieties. PS 1319 was characterized by cocoa beans rich in putrescine and spermidine, which could be detected in the cocoa seeds with a lower percentage of pulp. However, in Parazinho, the pulp was rich in spermidine and tyramine, whereas the seed contained moderate levels of tryptamine. 

During the fermentation of PS 1319, a larger number of amines were found; however, there was an increase in their levels, followed by a decrease, resulting in non-detectable levels at the end of fermentation. This was so for most of the amines: phenylethylamine (maximum levels at 96 h in E2); agmatine (maximum levels at 120–144 h, E1 and E3); spermidine (maximum at 72 h, E2); tyramine (maximum at 120 h, most treatments); and histamine (maximum at 144 h, E2). On the other hand, the levels of putrescine and cadaverine increased during fermentation. 

During the fermentation of Parazinho, there was an increase followed by a decrease in agmatine (maximum levels at 144 h). However, there were few changes in cadaverine and histamine levels. The levels of tyramine, putrescine, phenylethylamine, and tryptamine increased during fermentation.

At the end of fermentation, which took a longer time for PS 1319, this variety had only putrescine and cadaverine, which were already present in the original cocoa seeds, and tyramine, which was formed and accumulated during fermentation. Spermidine, which was present prior to fermentation, was degraded or used by yeast as a nitrogen source. The presence of cadaverine can suggest the presence of Enterobacteria, which have lysine decarboxylating enzymes. The formation and accumulation of tyramine suggest the occurrence of free tyrosine in the cocoa mass and the presence of microorganisms with tyrosine decarboxylation enzymes [4,11,20,25].

At the end of fermentation for the Parazinho variety, there was the formation and buildup of several amines, resulting in a fermented coca mass with five amines. This variety already had seven amines in the cocoa mass prior to fermentation, but, during fermentation, histamine, agmatine, and phenylethylamine were consumed, and cadaverine was formed and remained in the final product. The changes during processing were minor, except for tyramine, which was formed, and the levels increased 8- to 40-fold compared to initial levels. 

Summarizing, based on these results, the two varieties differed on their amine profile. The same tendency remained at the end of fermentation, with six amines in PS 1319 and five amines in Parazinho. In fermented PS 1319, there was a prevalence of cadaverine, whereas in Parazinho, there was a prevalence of tyramine. The accumulation of cadaverine is undesirable as it can impart a putrid flavor to the chocolate. The accumulation of tyramine would be undesirable for individuals taking monoaminoxidase inhibitor drugs and could lead to a hypertensive crisis [18].

The decrease in cocoa pulp during fermentation (from 100 to 0%) in PS 1319 only affected spermidine availability at the beginning of fermentation (increased levels) and tyramine levels at the end of fermentation (decreased levels). On the other hand, for Parazinho, the decrease in pulp availability at the beginning of fermentation affected a larger number of amines: it decreased the levels of agmatine, spermidine, and tyramine at time zero. But, at the end of fermentation, it resulted in higher levels of tyramine, spermidine, and cadaverine for 60% pulp treatment (E3). However, for the 0% pulp treatment (E4), no amine was present at the end of fermentation.

Putrescine is an intermediate amine in spermidine synthesis. Both play important roles in the development, growth, and survival of plants subject to sun, drought, heat, and oxidative stress. Serotonin participates in physiological processes such as reproduction, vegetative growth, melatonin production, germination, and bean longevity [29,30]. Note that cadaverine is undesirable, as studies have reported that it can lend putrefactive sensory characteristics to products [31].

The quantification of amines is important because histamine is primarily responsible for food poisoning, characterized by short incubation times (minutes to hours) and duration (hours). However, we must highlight that establishing toxicity levels for amines in food is complex as it depends on factors related to consumers, such as use of drugs, health status, and individual susceptibility [32,33].

We found, as in other studies on Brazilian cocoa fermentation, that spermidine is the predominant bioactive amine. We expected its presence since it occurs in living organisms in general and is related to cell growth, renewal, and metabolism. Moreover, not only did we expect it but also desired it as it is an effective and important antioxidant [1,11,17].

The literature has reported bioactive amines such as tyramine, phenylethylamine, tryptamine, serotonin, putrescine, and spermidine in cocoa during fermentation, including Brazilian ones. It should be mentioned that the composition of cocoa can be influenced by different conditions and regions of cultivation, degree of maturation, varieties, post-harvest handling, and storage. Thus, in fermentation, a series of microorganisms (such as bacteria and yeasts) associated with various chemical and biochemical reactions (such as protein hydrolysis and the formation of acids and ethanol) end up acting on these substances and result in the formation and release of various compounds. These compounds will also be impacted in other stages of the process, especially roasting. These new modifications that occur in the most diverse processing steps, all will affect the formation and degradation of bioactive amines [9,11,34]. 

Thus, fermentation favors the formation of amines. However, amines in fermented cocoa may also degrade since some microorganisms metabolize them [11,35]. Thus, the possibility of modulating the formation of bioactive amines in fermentative cocoa processes can be a promising tool to obtain high-quality chocolates with desirable functional and safety properties [9].

In general, during sun drying of the fermented cocoa, there was an increase in amine levels, except for a few amines. This is possible because dehydration is a lengthy process (up to 7 days) in which cocoa is kept at temperatures that allow microbial growth and enzyme activity, especially at the beginning of the process, when the water activity is high [36].

During roasting at 120 °C for 75 min, there was a decrease in the levels of putrescine, cadaverine, spermidine, and tryptamine. In fact, putrescine and cadaverine are volatile amines that can be lost during roasting. In most of the treatments, there was an increase in the levels of phenylethylamine and tyramine. This is probably due to the thermal decarboxylation of phenylalanine and tyrosine during cocoa roasting. It is interesting to observe that the increase in phenylethylamine was most evident in PS 1319, suggesting that this variety is richer in free phenylalanine compared to Parazinho. The enhanced levels of phenylethylamine in the roasted cocoa are important, as this amine is associated with mood modulation, and has beneficial effects on human health [34,37].

After roasting, PS 1319 had a wider range of amines (tyramine, spermidine, putrescine, cadaverine, and phenylethylamine) compared to Parazinho. This was especially true for the treatments with a higher pulp percentage. Delgado-Orpina and colleagues [25] found changes after roasting, including a greater content of some bioactive amines. Studies indicate that Strecker degradation forms these amines during the thermal decarboxylation of amino acids in the presence of α-dicarbonyl compounds formed during the Maillard reaction or lipid peroxidation [34,38,39].

Note that a decrease in bioactive compounds is normal at different stages of processing, affecting its final content and the functional properties of its derivatives [34,40]. However, the absence of amines such as agmatine in chocolate may be undesirable, as research claims it constitutes an important neurotransmitter in mammals since it can modulate physiological states related to pain and depression [41]. Amines such as spermidine (belonging to the polyamine group) are indispensable in all living cells, occurring at high levels in cell tissues with high growth rates [12,13].

As with liquor, phenylethylamine was the most abundant amine in chocolate. Since we found it in all assessed treatments, they seem to avoid the influence of pulp content and variety. Due to the association between phenylethylamine and aphrodisiac effects, mood improvement, and increased sensitivity, even low levels are desirable in cocoa and chocolate [1,18,42]. 

In general, the influence of the cocoa variety used for chocolate production seems to directly impact the amines. Moreover, some, such as putrescine, may be more susceptible to processing with total or partial pulp or without it. On the other hand, other amines, such as phenylethylamine, seem to resist the influence of variety or pulp content.

Parazinho showed bioactive amine values lower than those in PS 1319. By comparing chocolates produced and processed under the same conditions but with different varieties, the authors [20] report how genotypes influence the profile of bioactive amines. By comparing chocolate produced and processed under the same conditions [20], both bioactive amine types and levels significantly varied depending on genotypes. Other studies have found such variation. When they fermented Colombian criollo cocoa, they noticed a predominance of some amines before roasting, such as cadaverine, serotonin, histamine, and spermidine. They also found that temperature and roasting time can be important means to influence the amount and type of bioactive amines formed [25].

## 5. Conclusions

We conclude that processing influences the presence and amounts of bioactive amines in cocoa, whether in pre-processing, processing, or obtaining chocolate. Before processing, we only found putrescine (pulp) and serotonin (pulp and bean). During processing, other amines emerged, especially tyramine, tryptamine, putrescine, cadaverine, spermidine, and phenylethylamine. Of these, we found tyramine, putrescine, cadaverine, spermidine, and phenylethylamine in chocolate, especially in PS 1319. In general, Parazinho showed lower bioactive amine values than PS 1319, confirming the influence of variety on bioactive amine content. Finally, pulp was also important to determine the type and quantity of bioactive amines. In general, treatments with lower pulp content (especially without it) showed lower bioactive amine levels.

## Figures and Tables

**Figure 1 foods-12-00495-f001:**
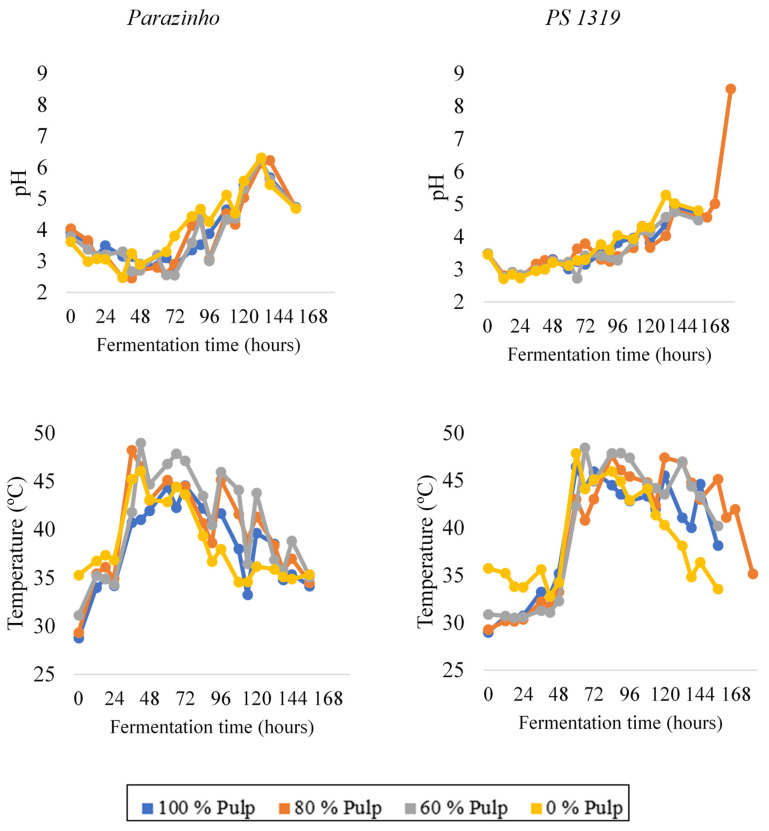
Temperature and pH changes during on-farm fermentation of two varieties of cocoa (PS 1319 and Parazinho) from Bahia, Brazil, using different proportions of pulp (100, 80, 60, and 0%).

**Figure 2 foods-12-00495-f002:**
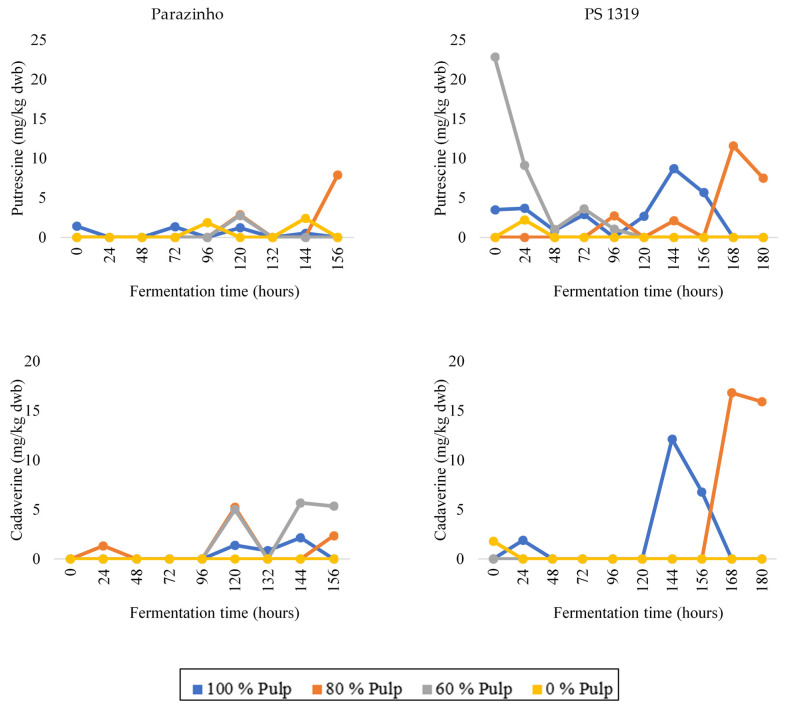
Changes in the levels of putrescine and cadaverine during on-farm fermentation of two varieties of cocoa (PS 1319 and Parazinho) from Bahia, Brazil, using different proportions of pulp (100, 80, 60, and 0%).

**Figure 3 foods-12-00495-f003:**
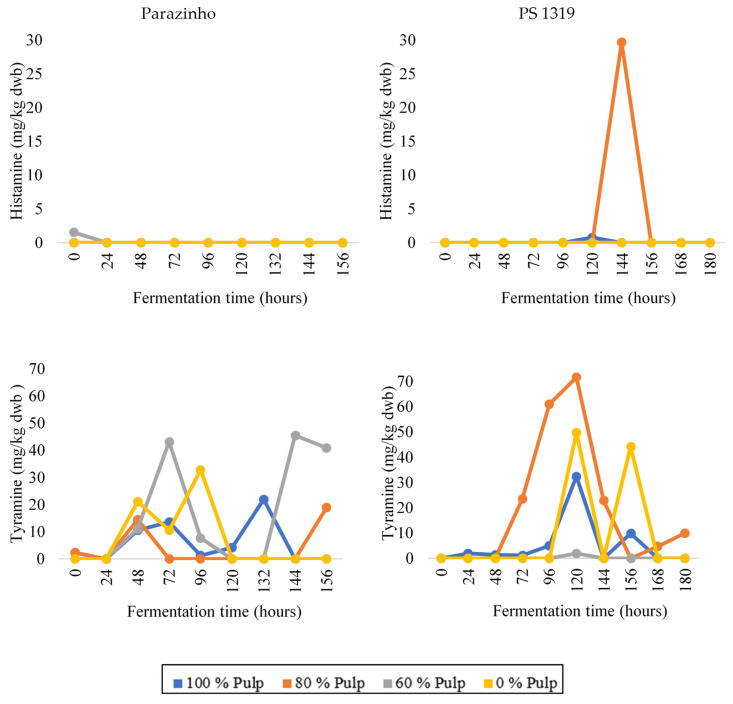
Changes in the levels of histamine and tyramine during on-farm fermentation of two varieties of cocoa (PS 1319 and Parazinho) from Bahia, Brazil, using different proportions of pulp (100, 80, 60, and 0%).

**Figure 4 foods-12-00495-f004:**
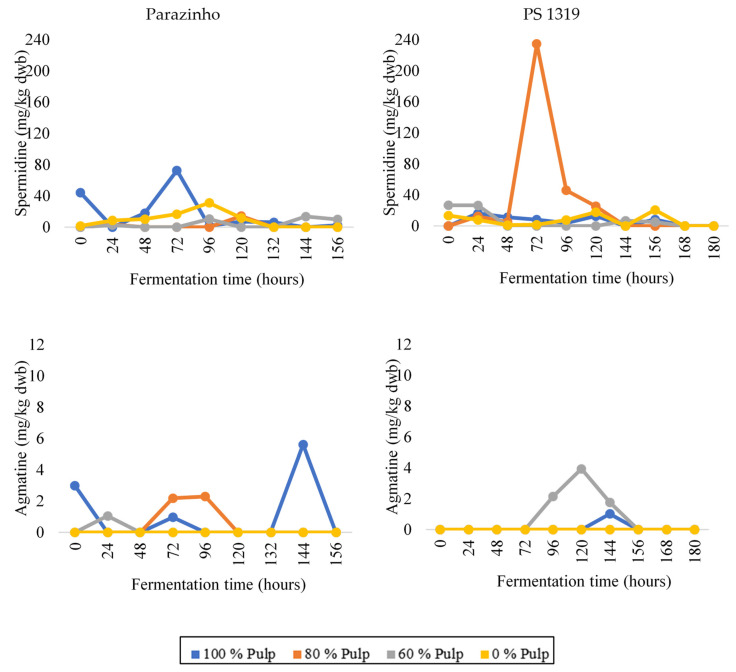
Changes in the levels of the polyamines—spermidine and agmatine—during on-farm fermentation of two varieties of cocoa (PS 1319 and Parazinho) from Bahia, Brazil, using different proportions of pulp (100, 80, 60, and 0%).

**Figure 5 foods-12-00495-f005:**
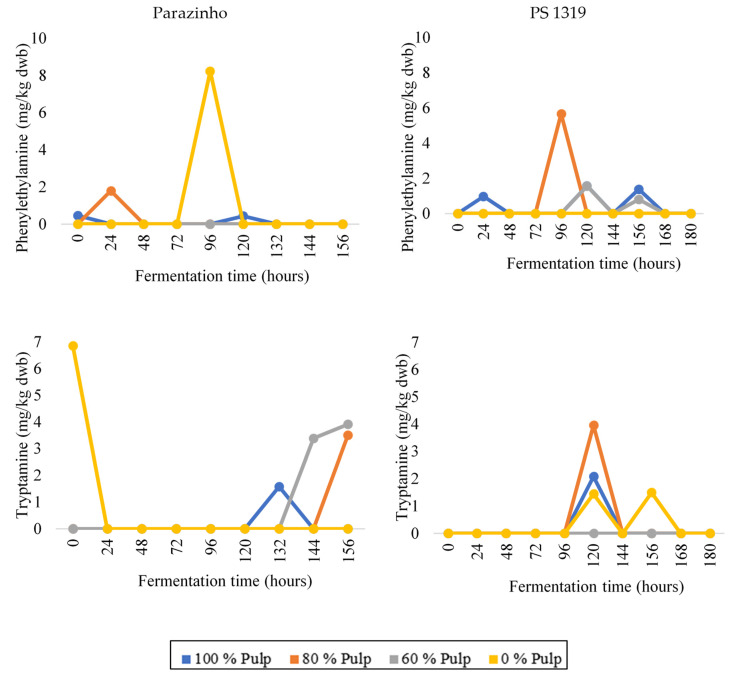
Changes in the levels of tryptamine and phenylethylamine during on-farm fermentation of two varieties of cocoa (PS 1319 and Parazinho) from Bahia, Brazil, using different proportions of pulp (100, 80, 60, and 0%).

**Figure 6 foods-12-00495-f006:**
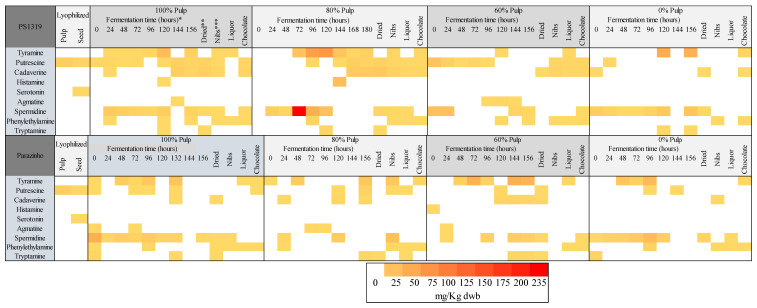
Heatmap representing free bioactive amine levels in cocoa and chocolate processing of two cacao varieties (PS 1319 and Parazinho) from Bahia, Brazil, using different pulp ratios (100, 80, 60, and 0%). * Fermentation time: cocoa seeds collected during the fermentation stage in different times. ** Dried: cocoa beans fermented and dried. *** Nibs: fermented and dried seed and roasted without the peel or germ.

**Figure 7 foods-12-00495-f007:**
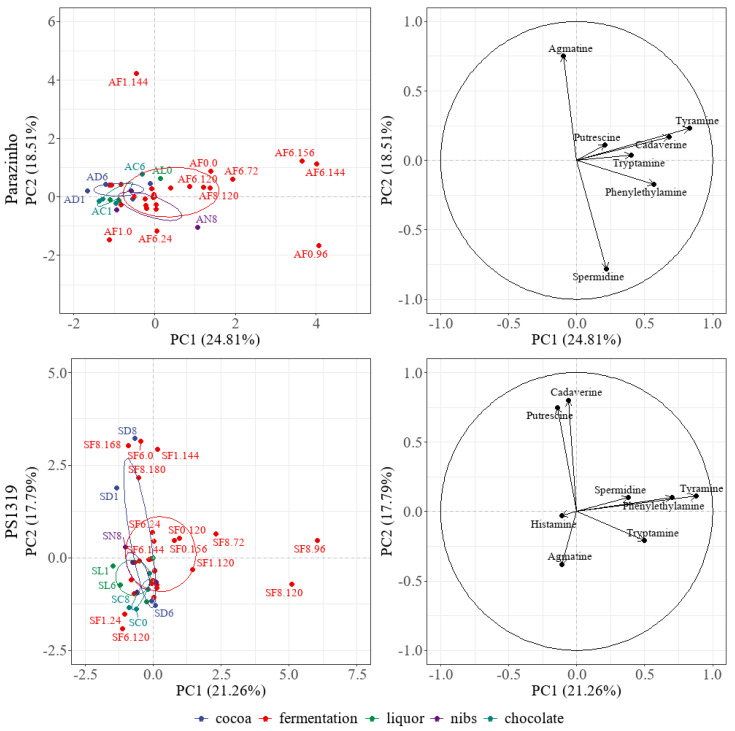
Principal Component Analysis of the bioactive amines during cocoa processing to compare Parazinho and PS 1319 varieties during processing steps. Note: S: PS 1319 variety. A: Parazinho variety. C: Control (freeze-dried cocoa pulp and seeds). F: fermented. D: Fermented and dried cocoa. N: Nibs (roasted cocoa). L: Liquor. C: Chocolate. 1: 100% pulp. 8: 80% pulp. 6: 60% pulp. 0: 0% pulp. F: Fermentation. 24, 48, 72, 96, 120, 144,156, 168, 180: fermentation times.

**Figure 8 foods-12-00495-f008:**
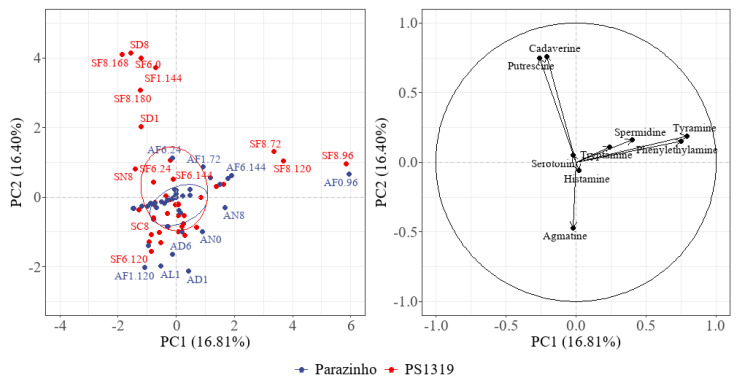
Principal Component Analysis of the bioactive amines during cocoa processing comparing Parazinho and PS 1319 varieties. Note: S: PS 1319 variety. A: Parazinho variety. C: Control (freeze-dried cocoa pulp and seeds). F: fermented. D: Fermented and dried cocoa. N: Nibs (roasted cocoa). L: Liquor. C Chocolate. 1: 100% pulp. 8: 80% pulp. 6: 60% pulp. 0: 0% pulp. F: Fermentation. 24, 48, 72, 96, 120, 144,156, 168, 180: fermentation times.

**Table 1 foods-12-00495-t001:** Levels of free bioactive amines in fermented and dried cocoa and in the nibs (roasted cocoa beans) of two varieties of cocoa (PS 1319 and Parazinho) from Bahia, Brazil, using different proportions of pulp (100, 80, 60, and 0%).

Variety	% Pulp	Cocoa	Biogenic Amines (mg/kg dwb)					
			Tym	Put	Cad	Spd	Phm	Trm
Parazinho	100	Fermented and dried	ND	0.23	0.61	3.71	ND	0.51
		Nibs (roasted) *	ND	ND	ND	ND	ND	ND
	80	Fermented and dried	ND	ND	ND	ND	ND	0.94
		Nibs (roasted)	ND	ND	ND	ND	ND	0.94
	60	Fermented and dried	ND	1.67	2.07	2.02	ND	1.04
		Nibs (roasted)	ND	ND	ND	ND	ND	ND
	0	Fermented and dried	ND	ND	ND	5.72	ND	ND
		Nibs (roasted)	ND	ND	0.66	ND	2.47	ND
PS 1319	100	Fermented and dried	ND	8.43	11.29	2.24	ND	1.10
		Nibs (roasted)	1.56	3.09	5.05	1.24	1.73	ND
	80	Fermented and dried	ND	10.48	17.96	3.38	ND	1.,79
		Nibs (roasted)	4.30	5.19	9.39	2.56	1.20	ND
	60	Fermented and dried	ND	ND	0.66	ND	ND	ND
		Nibs (roasted)	ND	3.47	5.48	ND	1.51	ND
	0	Fermented and dried	ND	ND	1.42	2.93	ND	ND
		Nibs (roasted)	ND	ND	ND	ND	0.57	ND

Tym: Tyramine. Put: Putrescine. Cad: Cadaverine. Spd: Spermidine. Phm: Phenylethylamine. Trm: Tryptamine. ND: < detection limit 0.40 mg/kg. dwb: dry weight basis. * Nibs: fermented and dried seed roasted without the peel or germ.

**Table 2 foods-12-00495-t002:** Levels of free bioactive amine in liquor and chocolate of two varieties of cocoa (PS 1319 and Parazinho) from Bahia, Brazil, using different proportions of pulp (100, 80, 60, and 0%).

Variety	% Pulp	Sample	Biogenic Amines (mg/kg dwb)					
			Tym	Put	Cad	Spd	Phm	Trm
Parazinho	100	Liquor	1.46	0.95	1.26	ND	0.95	ND
		Chocolate	2.32	ND	ND	ND	1.26	ND
	80	Liquor	ND	1.68	ND	ND	1.62	0.59
		Chocolate	3.39	ND	ND	ND	0.52	ND
	60	Liquor	3.88	ND	ND	ND	0.88	ND
		Chocolate	ND	ND	ND	1.16	1.49	ND
	0	Liquor	ND	ND	ND	0.86	2.60	ND
		Chocolate	2.49	ND	ND	ND	1.47	ND
PS 1319	100	Liquor	1.30	ND	ND	ND	0.64	ND
		Chocolate	ND	3.99	6.46	2.02	1.43	ND
	80	Liquor	ND	2.77	3.94	1.45	1.19	ND
		Chocolate	2.61	2.38	3.83	ND	1.08	ND
	60	Liquor	3.05	4.40	5.17	ND	0.77	ND
		Chocolate	ND	3.66	4.66	1.50	1.31	ND
	0	Liquor	ND	ND	ND	ND	ND	ND
		Chocolate	3.58	ND	0.62	ND	1.41	ND

Tym: Tyramine. Put: Putrescine. Cad: Cadaverine. Spd: Spermidine. Phm: Phenylethylamine. Trm: Tryptamine. ND: < detection limit 0.4 mg/kg. dwb: dry weight basis.

## Data Availability

Not applicable.

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
