# Peer review of "Varietal Influence on the Formation of Bioactive Amines during the Processing of Fermented Cocoa with Different Pulp Contents"

_foods, 2023, doi:10.3390/foods12030495_

Round 1
Reviewer 1 Report
The authors present an excellent work; it is well-written, the results are precise, and the conclusions agree with what was observed. Some aspects can be improved to give the reader a better understanding.
Line 103. Check the data. The volume declared does not fit the measurements presented. A hyphen is not placed between the number and the units.
3.3. Changes on bioactive amines throughout fermentation.
Please indicate in this stage and in subsequent steps if the content of biogenic amines corresponds to pulp, grain, or total. Before fermentation, the analysis was separated into pulp and grain (nibs), but it was not reported during fermentation. This is also not reported during drying, was the nib separated from the shell for analysis, or was the total analysis done? In roasting, it is reported that the shell was separated, were the amines in the shell measured?
Recommendation: Make a graph (heat map) showing the grain concentration of biogenic amines throughout the process.
Reviewer 2 Report
de Souza Silveira and co-authors studied the presence and formation of bioactive amines during the processing of two varieties of cocoa (PS1319 and Parazinho) with different proportions of pulp to obtain chocolates.
This project idea is of scientific interest and the experimental design is well structured.
The manuscript should be considered as major revision.
My suggestions are reported below:
- In several parts of the MS, there is a space after the dot as you can see in lines 41, 70, 71, 75 and so on.
- Quantitative results and not only qualitative results need to be reported in the abstract;
- It is necessary to explain the reasons of the choice of these analysis in the MS;
- Why did you choose to conduct the study on these two cocoa varieties? Explain it in the MS;
- Line 109: you indicate that the end of fermentation is based on color and aroma. They don't seem like two objective parameters to me. With what instruments are they evaluated? Is this procedure standardized? If yes, enter this information and references;
- Line 120: there are two commas;
- Lines 135-136: why did you choose these 9 amines? Enter the information in the MS.
- You should use the same value scale for y-axis to indicate the amount of amines (mg/kg dwb);
- In general, the discussion section needs to be written more thoroughly and critically;
- Lines 396-398: Learn more about this part. Explain how these factors affect the amine profile.
- In the introduction you write that amines can affect human health, giving several examples. In the light of the results obtained, describe what benefits there could be by consuming one variety of cocoa over the other.

Round 2
Reviewer 2 Report
my requests were met.